# Dr.Spider: A Diagnostic Evaluation Benchmark towards Text-to-SQL Robustness

**Shuaichen Chang**[1][*], **Jun Wang**[2], **Mingwen Dong**[2], **Lin Pan**[2], **Henghui Zhu**[2],
**Alexander Hanbo Li**[2], **Wuwei Lan**[2], **Sheng Zhang**[2], **Jiarong Jiang**[2], **Joseph Lilien**[2],
**Steve Ash**[2], **William Wang**[2], **Zhiguo Wang**[2], **Vittorio Castelli**[2], **Bing Xiang**[2], **Patrick Ng**[2]
[1] Ohio State University, [2] AWS AI Labs
chang.1692@osu.edu, {juwanga, zhiguow, patricng}@amazon.com

## Abstract

Neural text-to-SQL models have achieved remarkable performance in translating natural language questions into SQL queries. However, recent studies reveal that text-to-SQL models are vulnerable to task-specific perturbations. Previous curated robustness test sets usually focus on individual phenomena. In this paper, we propose a comprehensive robustness benchmark[1] based on Spider, a cross-domain text-to-SQL benchmark, to diagnose the model robustness. We design 17 perturbations on databases, natural language questions, and SQL queries to measure the robustness from different angles. In order to collect more diversified natural question perturbations, we utilize large pretrained language models (PLMs) to simulate human behaviors in creating natural questions. We conduct a diagnostic study of the state-of-the-art models on the robustness set. Experimental results reveal that even the most robust model suffers from a 14.0% performance drop overall and a 50.7% performance drop on the most challenging perturbation. We also present a breakdown analysis regarding text-to-SQL model designs and provide insights for improving model robustness.

## 1 Introduction

Large-scale cross-domain text-to-SQL datasets facilitate the study of machine learning models for generating a SQL query given a natural language question (NLQ) and corresponding database (DB) as input. Neural text-to-SQL models encode an NLQ and DB schema and decode the corresponding SQL (Wang et al., 2019; Lin et al., 2020; Scholak et al., 2021), which have achieved remarkable results on existing benchmarks (Zhong et al., 2017; Yu et al., 2018; Shi et al., 2020). However, those results are obtained in the setting where test data are created with the same distribution as training data. This setting prevents the evaluation of model robustness, especially when the data contain spurious patterns that do not exist in the wild. For example, previous studies (Suhr et al., 2020; Gan et al., 2021a; Deng et al., 2021) have found spurious patterns in the Spider (Yu et al., 2018), a widely used cross-domain text-to-SQL benchmark, such as NLQ tokens closely matching DB schemas, leading models to rely on lexical matching between NLQs and DB schemas for prediction instead of capturing the semantics that the task is intended to test.

Figure 1 shows examples where the state-of-the-art (SOTA) text-to-SQL models are vulnerable to perturbations. (1) DB perturbation: replacing the column name `winner_name` with `champ_name` leads the model to miss the intent of "3 youngest winners"; (2) NLQ perturbation: the model confuses the selected column `winner_name` with `winner_age` given a paraphrased NLQ which uses "Who" to imply the selected column; (3) SQL perturbation: a simple change to the number of returned items (from `LIMIT 3` to `LIMIT 8`) fails the model to detect the right intent. Recent studies created data to reveal the robustness problem of text-to-SQL models via perturbing DBs or NLQs (Ma & Wang, 2021; Pi et al., 2022; Gan et al., 2021a; Deng et al., 2021). However, they usually focus on individual linguistic phenomena and rely on rule-based methods or a few

---

[*]Work done during internship at AWS AI Labs
[1]Our data and code are available at `https://github.com/awslabs/diagnostic-robustness-text-to-sql`.

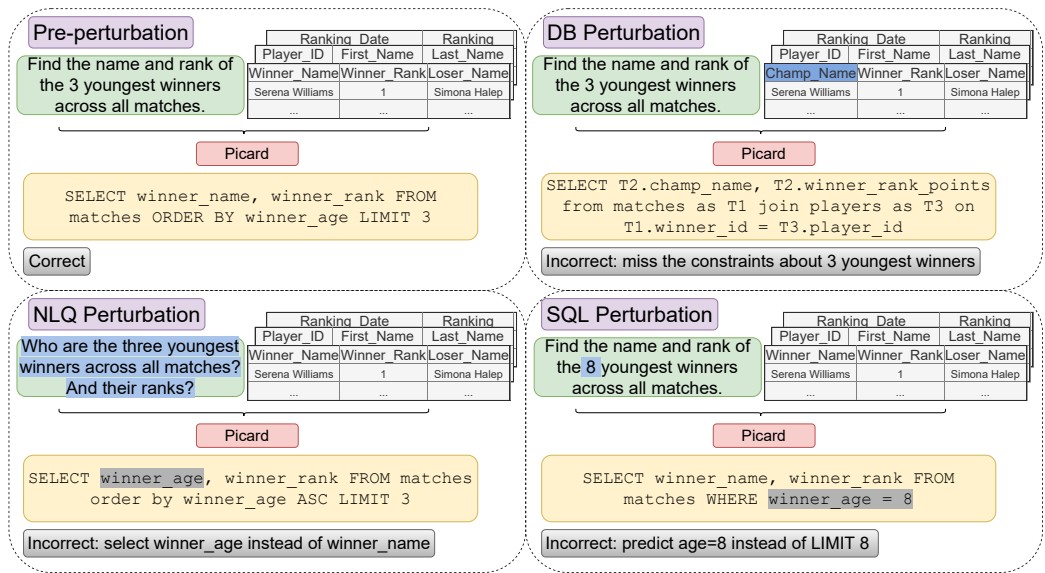

Figure 1: An example of the SOTA model Picard (Scholak et al., 2021) against DB, NLQ, SQL perturbations on the database WTA. Picard predicts a correct SQL on pre-perturbation data but fails on post-perturbation data. The blue and gray areas highlight the modification on input and the errors of predicted SQLs respectively.

annotators, which cannot cover the richness and diversity of human language. For example, the NLQ perturbation example in Figure 1 requires sentence-level paraphrasing, which cannot be generated by previous perturbation methods.

In this paper, we curate a comprehensive **D**iagnostic **R**obustness evaluation benchmark (**Dr.Spider**), based on Spider (Yu et al., 2018) via 17 perturbations to cover all three types of robustness phenomena on DBs, NLQs, and SQLs. Dr. Spider contains 15K perturbed examples. We perturb DBs with a set of predefined rules by taking advantage of their structural nature to represent data in different ways. For NLQ perturbations, we propose a collaborative expert-crowdsourcer-AI framework by prompting the pretrained OPT model (Zhang et al., 2022) to simulate various and task-specific linguistic phenomena. For SQL perturbations, we programmatically modify the local semantics in SQLs and their corresponding tokens in NLQs while minimizing the surface-level changes to measure model robustness to local semantic changes.

We evaluate the SOTA text-to-SQL models on our benchmark (Wang et al., 2019; Rubin & Berant, 2021; Yu et al., 2021; Scholak et al., 2021). The experiments demonstrate that although the SOTA models achieve good performance on the original data, they struggle to have consistently correct predictions in our robustness sets. Even the most robust model suffers from a 14.0% performance drop overall and a 50.7% performance drop against the most challenging perturbation. We also present a breakdown analysis of model robustness in terms of model architectures, including model size, decoder architecture, and the entity-linking component. We analyze the advantage of different model designs, which provides insights for developing robust text-to-SQL models in the future.

## 2 RELATED WORK

**Robustness in NLP** The evaluation of model robustness is an important step toward the development of reliable models. (Nie et al., 2019; Wang et al., 2021; Goel et al., 2021). Jia & Liang (2017); Iyyer et al. (2018); Wang et al. (2021) measure model robustness against semantic-preserving perturbations, and Kaushik et al. (2019); Gardner et al. (2020) evaluate the decision boundaries of models against semantic-changing perturbations, which change local semantics while minimizing modifications in the surface-level patterns. Previous studies on robustness data for text-to-SQL only consider semantic-preserving perturbations on DB or NLQ and rely on rule-based methods or handcrafted examples, which limits the naturalness and diversity of perturbations (Gan et al., 2021a;b; Deng

et al., 2021; Pi et al., 2022). Our benchmark not only contains semantic-preserving perturbations (on DB and NLQ) but also semantic-changing perturbations (on SQL) to evaluate the robustness of models in distinguishing closely related but different semantics.

**Data Generation with Pre-trained Language Models**   Large pre-trained language models (PLM), such as GPT-3 (Brown et al., 2020), OPT (Zhang et al., 2022), PaLM (Chowdhery et al., 2022), have achieved human-level performance on many NLP tasks. Recently, they have been used to generate data with prompts that consist of a few similar examples (Schick & Schütze, 2021; Lee et al., 2021b; 2022; Liu et al., 2022). Liu et al. (2022) has motivations that most parallel ours, which uses GPT-3 to generate a new data example with several existing data examples that share a similar pattern. However, instead of generating a new example, we leverage PLMs to paraphrase existing NLQs to create task-specific perturbations.

## 3   DATA CREATION

Dr.Spider aims to comprehensively evaluate the robustness of models with perturbations on each component of the text-to-SQL task, i.e. DB, NLQ, and SQL. We apply task-specific perturbations to create our benchmark based on the Spider development set, as the Spider test set is not public. Dr.Spider contains 3 DB perturbation test sets, 9 NLQ perturbation test sets, and 5 SQL perturbation test sets to simulate various task-specific phenomena. Each test set contains parallel pre-perturbation and post-perturbation data to measure model robustness against the perturbation. In total, Dr.Spider contains 15K pairs of pre-perturbation and post-perturbation examples.

**Data Creation Principles**   We create our data following three principles. **Task Specificity:** Huang et al. (2021); Ma & Wang (2021) uses general perturbations in NLQs, such as word substitution and back translation, making it difficult to provide fine-grained analysis in addition to the general robustness problems found in other NLP tasks (Nie et al., 2019; Ribeiro et al., 2020; Goel et al., 2021). We curate our NLQ perturbation data within 9 task-specific categories to reveal unique robustness challenges in the text-to-SQL task. **Linguistic Richness:** Previous studies rely on rule-based word-level perturbations (Gan et al., 2021a; Ma & Wang, 2021) or manual perturbations provided by a few experts (Gan et al., 2021b; Deng et al., 2021) which cannot provide a high degree of linguistic richness. For example, "teachers whose hometown are" is perturbed to "teachers whose birthplace are" with word-level perturbation (Gan et al., 2021a), however, other common linguistic phenomena are not covered in previous data such as the sentence-level perturbation "teachers who were born in" from Dr.Spider. We leverage large pre-trained language models (PLMs) for NLQ perturbations instead of relying on rule-based methods or a few human annotators. **Diagnostic Comprehensiveness:** For a systematic evaluation, a benchmark should provide various diagnostic angles. Previous work only considers semantic-preserving perturbations on either DBs or NLQs with a few perturbation methods, such as replacing the database schema with synonym (Pi et al., 2022) and replacing schema indicator in NLQs with synonyms Gan et al. (2021a). We apply various perturbations, including semantic-preserving perturbations to DBs and NLQs and semantic-changing perturbations to SQLs, to curate 17 perturbation test sets focusing on different robustness phenomena.

### 3.1   DATABASE PERTURBATION

Database perturbation aims to simulate the scenario where data can be represented in different ways in a DB. We consider three DB perturbations: `schema-synonym`, `schema-abbreviation`, and `column-equivalence`. **Schema-synonym** replaces the name of a column with its synonym. For instance, the column named `country` can be replaced with `nation`. **Schema-abbreviation** replaces the name of a column with its abbreviation. For example, the column named `ranking_points` can be substituted with `rank_pts`. First, we create a schema synonym and an abbreviation dictionary for each database. We use the synonym dictionary in Pi et al. (2022) and obtain an abbreviation dictionary from a public abbreviation website [2]. Unlike Pi et al. (2022) who replace all possible columns with their synonyms, we sample a subset from the synonym and abbreviation dictionary to replace the sampled columns with their synonyms or abbreviations. The samplings are repeated five times to create diverse perturbations with duplicate samples removed. Finally, we programmatically modify the database and affected SQLs to generate

---

[2]https://www.allacronyms.com/

new data based on sampled perturbations each time. As Ma & Wang (2021) discovered that models are generally robust to DB perturbations that do not affect gold SQLs , we only select SQLs that mention any replaced columns in perturbations.

**DBcontent-equivalence** simulates the scenario that data can be represented in different formats in a database by replacing a column (column name and content) with one or multiple semantic-equivalent columns. We first ask task experts to provide possible semantically equivalent substitutions for each column. (1) For text column content, we consider to split a compound column into multiple columns, such as replacing `fullname` with `firstname` and `lastname`. (2) We convert columns with fixed string values into boolean column(s) or vice versa. For instance, a boolean column `is_male` is equivalent to a text column `sex` with all gender options, and a text column `degree` is equivalent to three boolean columns `is_bachelor`, `is_master`, and `is_PhD` when there are only three degrees in the database. (3) For numerical columns, we replace them with other numerical columns with equivalent semantics. For example, the column `birthyear` is semantically equivalent to the column `age` since one can be inferred from the other. Similar to the other DB perturbations, we sample a subset from the substitutions at most 5 times to ensure perturbation diversity and programmatically modify SQLs according to the changes in DBs. Note that we remove SQLs that require unseen grammar to modify according to the database perturbation. To the best of our knowledge, we are the first to consider representing database content in different ways to create database perturbations.

## 3.2 NATURAL LANGUAGE QUESTION PERTURBATION

We propose a collaborative expert-crowdsourcer-AI framework to collect task-specific perturbations by leveraging the advantage of each: experts (we) provide task-specific analysis, crowdsourcers have natural and diverse language patterns, and AI models are scalable. Our framework involves the following phases: (1) Crowdsourcing annotators from Amazon Mechanical Turk [3] paraphrase sampled questions from the Spider dataset. (2) Task experts review the paraphrases and categorize them from the text-to-SQL task perspectives. (3) A pretrained language model (PLM) generates categorized paraphrase perturbations based on the collected human paraphrases in each category. (4) A pretrained natural language inference (NLI) model and rule-based filters are used to select factual paraphrases in each category. (5) Task experts review the paraphrased questions in the end to ensure the high quality of our dataset.

**Crowdsourcing Paraphrase Study** NLQs in Spider were collected in a setting that users (questioners) are familiar with SQL and have the access to the DB schema and content. However, in reality, the general users are not familiar with DBs or SQLs in most applications (Lee et al., 2021a). To understand the linguistic phenomena that NLQs could have in a realistic scenario, we recruit annotators on Amazon Mechanical Turk to paraphrase 100 sampled questions in Spider and collect five paraphrases per question. We encourage annotators to create diverse paraphrases by not directly providing the database schema or content to the annotators. Instead, we provide a short description of each database, which reveals the topic of the database and the key tables and columns in the database. Note that we omit the description of naturally occurring tables and columns (e.g. a `student` table contains columns `name`, `grade`) because they do not provide the necessary information for understanding the database context and also make annotators tend to follow the structure of schema during paraphrasing.

**Expert Analysis** We find only 270 out of 500 paraphrases are factual as a small change to a question could alter its meaning in the text-to-SQL task. We remove nonfactual paraphrases and analyze linguistic phenomena in crowdsourcing paraphrases regarding how the paraphrase is different from the original question. We focus on three types of tokens in NLQ that can be aligned with SQL: SQL keyword indicators, column name indicators, and value indicators, where SQL keywords include sort keywords, comparison, and aggregation operators; column names refer to the mentioned column in SQL except for `JOIN` clause, and value tokens include database content and other numbers.

We find 9 categories based on whether the paraphrasing contains a task-specific linguistic phenomenon, including 8 categories that modify the aforementioned indicators and one category that does not. Table 1 shows the categorized examples. Note that a paraphrase may fall into multiple categories if it contains multiple phenomena. The category `Multitype` also include such examples.

---

[3]https://www.mturk.com/

We believe that the first 8 categories can diagnose models against each task-specific NLQ perturbation, and we regard the last category as a control group to evaluate the robustness of models against general syntactic perturbations in sentence-level paraphrasing.

| Paraphrase Category | Example |
|---|---|
| `Keyword-synonym` (Replace SQL keyword indicators with synonyms) | Question: What is the code of airport that has the highest number of flights?
Paraphrases: Show me the code for the airport that currently has the most flights. |
| `Keyword-carrier` (Imply SQL keyword indicators by carrier phrases) | Question: Show the name and theme for all concerts and the number of singers in each concert.
Paraphrase: List the names and themes for all concerts and how many singers are in each. |
| `Column-synonym` (Replace column indicators with synonyms) | Question: List the name of teachers whose hometown is not Little Lever Urban District.
Paraphrases: Find the name of teachers who were not born in Little Lever Urban District. |
| `Column-carrier` (Imply column indicators by carrier phrases) | Question: Show the name of teachers aged either 32 or 33?
Paraphrases: Which teachers are aged either 32 or 33. |
| `Column-attribute` (Imply column indicators by aggregated attributes) | Question: What is the name of the conductor who has worked the greatest number of year?
Paraphrases: Who has worked the longest as conductor? |
| `Column-value` (Imply column indicators by values) | Question: What are the ids of the students who do not own cats as pets?
Paraphrases: Find the IDs of students who don't own cats. |
| `Value-synonym` (Replace value indicators with synonyms) | Question: Find all airlines that have at least 10 flights.
Paraphrases: Show the number of airlines that have at least ten flights. |
| `Multitype` (Contain multiple phenomena above) | Question: Find number of pets owned by students who are older than 20?
Paraphrases: How many pets are owned by students over 20? |
| `Others` | Question: what is the name and nation of the singer who have a song having 'Hey' in its name?
Paraphrases: Which singers have 'Hey' in their song's name? List their name and nation. |

Table 1: Crowdsouring paraphrase categories and examples. The red tokens in the original questions highlight the keyword/column/value indicators that are paraphrased, and the blue tokens in the paraphrases represent how the indicators are replaced or implicitly mentioned.

**Categorized Paraphrase Generation**    Although crowdsourcing paraphrases can be used as NLQ perturbations, it is costly to scale and categorize them for curating a large-scale dataset. To tackle this issue, we generate paraphrases with large PLM and prompt it with crowdsourcing paraphrases in each category. We choose the OPT (Zhang et al., 2022) with 66B parameters as the PLM model. We manually select five examples from the crowdsourcing paraphrases in each category to form the in-context examples as the prompt prefix.

For each category, the input of OPT consists of a category-specific instruction, 5 question-paraphrase pairs in the category, an NLQ that needs to be paraphrased, and its SQL tokens (keywords/columns/values) whose indicators are required to be modified in the category. Inspired by Wei et al. (2022); Paranjape et al. (2021), we employ the OPT model to generate an explanation along with the paraphrase. We overgenerate 20 paraphrases per question to create diverse paraphrases (however, at most 5 examples will be kept after the filtering stage). Our prompt prefixes and implementation details can be found in Appendix A.2.

**Paraphrase Filtering and Quality Control**    After generating paraphrases in each category, we first remove the duplicate questions and use a BART-large model (Lewis et al., 2019) trained on Multi-NLI (Williams et al., 2018) to filter out nonfactual paraphrases. Then, a simple and high-recall rule-based classifier is built to remove out-of-category paraphrases based on the alignment between SQL tokens and their indicators in NLQ. In particular, we use the annotations from Lei

et al. (2020) to build the column and value indicators. And to obtain SQL keyword indicators, we collect the common indicators (covering over 80% examples) of each SQL keyword in the NLQs and programmatically label NLQs with those indicators. The common indicators of SQL keywords can be found in Appendix A.3. Finally, we filter out the paraphrases where all the indicators for SQL keywords, columns and values remain the same as the original NLQ except category `others`.

To ensure the data quality in Dr.Spider, we request three annotators with SQL expertise to carefully review paraphrases with gold SQL labels and select the final data. We split the filtered data into small chunks and each chunk is reviewed by at least one annotator. 63% generated paraphrases are labeled factual and belong to the correct category, while only 54% of human paraphrases from the previous crowdsourcing are actually factual. This shows the effectiveness of our proposed framework for generating categorized paraphrases. 5% data (evenly from 9 categories) are reviewed by all three annotators. The Fleiss Kappa score between the three annotators is 0.61. Randomly sampled generated paraphrases in each category can be found in Appendix A.3.

## 3.3 SQL PERTURBATION

SQL perturbations evaluate models' robustness against local semantic changes. We consider five perturbations regarding SQL tokens: (1) **Comparison** replaces a comparison operation from $\{<, >, <=, >=\}$ to another; (2) **Sort-order** switches ascending sort and descending sort; (3) **NonDB-number** replaces a non-DB number $n$ with another number in $[n - 10, n + 10]$, including the number in the `LIMIT` clause (e.g. `Limit n`) and a criteria about counts (e.g. `HAVING count(*) > n`); (4) **DB-text** replaces a mentioned DB content text to another in the same column; (5) **DB-number** replaces a mentioned DB content number $m$ to another number in $[m - 10, m + 10]$. Similar to DB and NLQ perturbations, we perturb an example at most five times to increase the diversity of perturbations. Examples for each category can be found in Appendix A.4

Since SQL perturbations involve modifications on both SQL tokens and their indicators in NLQs, we want to minimize the surface-level changes introduced to NLQs to disentangle SQL perturbations from NLQ perturbations. For `comparison` and `sort-order`, we replace an indicator in NLQ only with common indicators in the Spider training data. For example, an NLQ "...*more than one car*" corresponds to the SQL "`...count(*)>1`". We perturb the SQL by replacing operation ">" with ">=" as well as the NLQ indicator "more than" with "at least". We choose "at least" rather than uncommon indicator phrases (e.g. "no less than") because SQL perturbations focus on evaluating models against local semantic changes on SQLs and replacing the SQL keyword indicator "at least" with its synonym "no less than" is covered in NLQ perturbation `keyword-synonym`. More details about the replacement rules can be found in Appendix A.4. For number perturbations, a mentioned number in NLQ will only be replaced with another number in the same format, e.g., 3=>4, 3rd=>4th, and three=>four, and we skip numbers 0 and 1 since they usually have non-number tokens as indicators in NLQ. We only consider the DB content text that is mentioned in the exact same format in NLQ to disentangle it from NLQ perturbation `value-synonym`.

## 3.4 DATA STATISTICS

| Perturbation Data | TS | LR | DC | # DB Perturbations | # NLQ Perturbations | # SQL Perturbations |
|---|---|---|---|---|---|---|
| Huang et al. (2021) | × | ✓ | ✓ | - | 6 | - |
| Ma & Wang (2021) | ✓ | × | ✓ | 8 | 4 | - |
| Pi et al. (2022) | ✓ | × | × | 2 | - | - |
| Gan et al. (2021a) | ✓ | × | × | - | 1 | - |
| Gan et al. (2021b) | ✓ | × | × | - | 5 | - |
| Deng et al. (2021) | ✓ | × | × | - | 1 | - |
| **Dr.Spider** | ✓ | ✓ | ✓ | 3 | 9 | 5 |

Table 2: Perturbation attribute and the number of perturbation types for DB, NLQ, and SQL in existing studies and Dr.Spider. TS, LR, and DC represent task-specificity, linguistic richness, and diagnostic comprehensiveness, respectively.

Table 2 shows the covered principles and the number of perturbation types for DB, NLQ, and SQL in previous work and ours. Dr.Spider is the only benchmark covering all principles and perturbing

all DB, NLQ, and SQL. Dr.Spider also contains various task-specific paraphrases for NLQ perturbations which contain a higher degree of linguistic richness than word-level perturbation methods. Similar to ours, Ma & Wang (2021) consider various formats to represent data in a DB. However, their perturbations require NLQ and SQL not to mention the perturbed DB columns, which fail to detect the lack of robustness of models (2.5 points performance drops on average). We report the number of examples in Dr.Spider and the original Spider development set in the column # in Table 3. The number represents the size of each post-perturbation test data in Dr.Spider. To measure the naturalness of our generated NLQ perturbations, we sample 500 questions from each of the original Spider development set, human paraphrases (we use all 270 factual paraphrases), and our perturbed questions. Crowdsourcing annotators are hired to score the clarity and fluency of sampled questions from 1 to 3. Each question is evaluated by five annotators. The clarity scores of Spider questions, human paraphrases, and generated paraphrases are 2.7, 2.6, 2.6, and fluency scores are 2.7, 2.6, 2.7, which shows generated paraphrases have similar fluency and clarity to the original questions and human paraphrases. The annotation interface can be found in Appendix A.3.

## 4 EXPERIMENTS

**Models**   The SOTA text-to-SQL models follow an encoder-decoder framework that encodes an NLQ and a DB schema jointly and decodes the corresponding SQL. We evaluate multiple representative text-to-SQL models on Dr.Spider, which are trained on the Spider training set: (1) **RATSQL** (Wang et al., 2019): The encoder is pretrained with BERT-large (Devlin et al., 2018) and augmented with relation-aware attention (Shaw et al., 2018), and the decoder is a top-down docoding with abstract syntax tree (AST). We use the implementation from Scholak et al. (2020). (2) **GRAPPA** (Yu et al., 2021): RatSQL with GraPPa embedding as pretrained encoder. GraPPa finetunes RoBERTa-large (Liu et al., 2019) on synthetic table-text pair data. (3) **SMBOP** (Rubin & Berant, 2021): An encoder-decoder model with the same encoder as GRAPPA and a bottom-up decoder with AST. (4) **T5-FAMILY** (Raffel et al., 2020): We use T5-BASE, T5-LARGE, and T5-3B which are fine-tuned on Spider. (5) **T5-3B LK** (Shaw et al., 2021): A T5-3B model with an entity linking between question and DB content. (6) **PICARD** Scholak et al. (2021): T5-3B LK with a constraint decoding. It is the SOTA model on the Spider benchmark. (7) **GPT-3 CODEX** Chen et al. (2021): A large language model without finetuning on Spider. We present the evaluation of CODEX in Appendix B.

**Robustness Evaluation Metrics**   We follow the Spider Benchmark to use the two SQL evaluation metrics to evaluate the correctness of a predicted SQL with a gold SQL: exact set match (EM) measures a predicted SQL with a gold SQL on each SQL clause ignoring values; execution accuracy (EX) compares the denotation answers from a predicted SQL and a gold SQL. We use EX as our main evaluation metric as it evaluates the correctness of SQL values, which is an important part of model robustness, while we only report EM for GRAPPA which omits value predictions.

To evaluate robustness, we report 3 metrics: (1) **pre-perturbation accuracy**: the accuracy on pre-perturbation data, (2) **post-perturbation accuracy** (or absolute robustness accuracy): the accuracy on post-perturbation data, (3) **relative robustness accuracy**: The ratio of the number of correct predictions on both parallel pre-perturbation and post-perturbation data over the number of correct predictions on pre-perturbation data. While the post-perturbation accuracy measures the absolute performance of a model on perturbation data, relative robustness accuracy is to evaluate the regression rate of models against perturbation, which is more precise to compare the robustness of models by considering their performances on the pre-perturbation data.

## 5 RESULTS

### 5.1 DIAGNOSTIC RESULTS

Table 3 shows the execution accuracy changes of the SOTA models against perturbations, where we report the macro-average scores for all perturbations. The EM accuracy can be found in Appendix B. In general, all models suffer from significant performance drops in DB and NLQ perturbations. `Picard` is the most robust model but still has 30.0% and 14.5% relative performance drops. The most challenging perturbation for DB is `DBcontent-equivalence`. Even PICARD faces a 50.7% performance drop. We believe it is because the way that schemas and contents are mentioned in NLQs is different from that in DB, which requires additional common-sense reasoning to align

| | Perturbation | # | RATSQL | SMBOP | T5-BASE | T5-LARGE | T5-3B | T5-3B LK | PICARD |
|---|---|---|---|---|---|---|---|---|---|
| | Spider-dev | 1,034 | 72.8* | 78.0 | 57.0* | 67.2 | 71.7* | 74.4 | **79.3** |
| DB | Schema-synonym | 2,619 | 65.7—45.4 | 70.8—53.9 | 48.8—25.2 | 59.4—36.8 | 64.2—41.6 | 66.4—46.9 | **73.0**—**56.5** |
| | Schema-abbreviation | 2,853 | 67.0—44.2 | 72.2—59.0 | 48.5—25.6 | 61.2—39.2 | 66.0—50.7 | 69.5—53.3 | **74.9**—**64.7** |
| | DBcontent-equivalence | 382 | 79.8—12.0 | 81.2—37.2 | 56.0—17.5 | 71.5—34.0 | 78.3—36.4 | 84.6—40.8 | **88.7**—**43.7** |
| | Avergae | - | 70.8—33.9 | 74.7—50.0 | 51.1—22.8 | 64.0—36.7 | 69.5—42.9 | 73.5—47.0 | **78.9**—**55.0** |
| NLQ | Keyword-synonym | 953 | 69.9—53.7 | **75.9**—64.3 | 53.9—45.8 | 66.5—57.7 | 68.3—60.3 | 70.2—62.6 | 72.6—**66.3** |
| | Keyword-carrier | 399 | 83.7—81.0 | 82.7—79.2 | 68.9—66.2 | 74.9—70.9 | 82.2—76.9 | 82.7—76.4 | **85.0**—**82.7** |
| | Column-synonym | 563 | 63.8—42.6 | 65.9—48.7 | 45.3—28.6 | 54.5—40.1 | 63.1—46.5 | 63.9—51.3 | **71.0**—**57.2** |
| | Column-carrier | 579 | 79.8—58.0 | 85.1—64.6 | 57.3—40.8 | 72.9—57.3 | 76.7—59.6 | 83.1—61.7 | **86.9**—**64.9** |
| | Column-attribute | 119 | 53.8—42.9 | **75.6**—**58.0** | 52.1—40.3 | 47.9—42.9 | 53.8—52.1 | 49.6—48.7 | 58.8—56.3 |
| | Column-value | 304 | 73.7—52.6 | 79.6—58.9 | 52.6—30.3 | 64.1—45.4 | 66.1—50.0 | 69.1—58.6 | **82.9**—**69.4** |
| | Value-synonym | 506 | 62.3—18.6 | 68.6—29.1 | 45.8—26.3 | 60.5—36.6 | 64.0—35.8 | 68.6—46.4 | **72.5**—**53.0** |
| | Multitype | 1,351 | 70.2—39.7 | **76.2**—46.1 | 51.4—33.0 | 63.8—44.0 | 67.8—47.0 | 70.1—51.1 | 74.4—**57.1** |
| | Others | 2,819 | 74.5—67.2 | 79.4—73.7 | 58.0—51.3 | 67.6—62.9 | 71.4—66.0 | 75.3—73.1 | **79.6**—**78.3** |
| | Average | - | 70.2—50.7 | **76.6**—58.1 | 53.9—40.3 | 63.6—50.9 | 68.2—54.9 | 70.3—58.9 | 76.0—**65.0** |
| SQL | Comparison | 178 | 62.9—59.6 | **68.5**—65.2 | 50.0—32.6 | 57.3—57.3 | 65.2—60.1 | 62.9—62.4 | 68.0—**68.0** |
| | Sort-order | 192 | 72.9—68.2 | 75.5—**76.6** | 62.0—60.4 | 74.5—68.8 | 74.5—73.4 | 75.0—70.3 | **79.2**—74.5 |
| | NonDB-number | 131 | 77.9—58.8 | 77.1—71.8 | 66.4—62.6 | 74.8—73.3 | 81.7—**82.4** | 77.1—73.3 | **83.2**—77.1 |
| | DB-text | 911 | 57.5—51.2 | 65.1—63.1 | 38.1—38.1 | 44.0—48.0 | 53.0—52.1 | 59.5—58.3 | **64.7**—**65.1** |
| | DB-number | 410 | 72.9—74.4 | **87.1**—84.4 | 68.0—65.4 | 76.8—76.3 | 80.0—79.3 | 83.9—83.7 | 86.3—**85.1** |
| | Average | - | 68.8—62.4 | 74.7—72.2 | 56.9—51.8 | 65.5—64.7 | 70.9—69.5 | 71.7—69.6 | **76.3**—**74.0** |
| All | | - | 69.9—51.2 | 75.7—60.8 | 54.3—40.6 | 64.2—52.4 | 69.2—57.1 | 71.3—59.9 | **76.6**—**65.9** |

Table 3: Data statistics and the execution (EX) accuracy of SOTA text-to-SQL models on the original Spider development set (Spider-dev) and Dr.Spider. The column # contains the size of Spider-dev and post-perturbation data in each perturbation test set in Dr.Spider. * represents the model we trained with official codes as their models are not public. x—y represents the pre-perturbation accuracy and post-perturbation accuracy (i.e. absolute robustness accuracy). We report macro-average scores over multiple perturbations. Bold number is the highest accuracy in each category and the underscore stands for the second best result.

them. This phenomenon is common in reality when the actual users are not aware of DB schemas. Prediction samples of PICARD can be found in Appendix B.

Although all NLQ perturbations are based on sentence-level paraphrasing, models are more vulnerable to perturbations that contain task-specific phenomena than `others`, which demonstrates the value of our perturbation categories. PICARD is more vulnerable to replacing keyword indicators with synonyms than question carriers. For column-related perturbations, PICARD is more vulnerable to the perturbation that implies columns with question carriers or values than replacing column names with synonyms. The most challenging perturbation among NLQ is the `value-synonym` where PICARD has a 26.9% performance drop. Since most SOTA models rely on string matching to align the DB value with its indicator in NLQ, this perturbation illustrates the limits of such methods when values are represented in different formats.

Surprisingly, although most SOTA models are relatively robust to local semantic changes (SQL perturbations), except for RATSQL and T5-BASE, PICARD still has a 7.3% performance drop in `NonDB-number`. Figure 1 contains an example that PICARD understands "*3 youngest winners*", but confuses "*8 youngest winners*" with "*age is 8*". A robust model should recognize that the number in the question corresponds to a logical meaning `LIMIT BY` and be robust to the changes in the number value.

## 5.2 DIAGNOSTIC INSIGHT FOR MODEL DESIGNS

This section presents experiments to answer some questions about the robustness in terms of model designs and provide insights for developing more robust models.

**Are larger models more robust?** Figure 2 shows the results of T5-FAMILY models. Not surprisingly, the larger models have a better pre-perturbation accuracy. In addition, we find that the larger

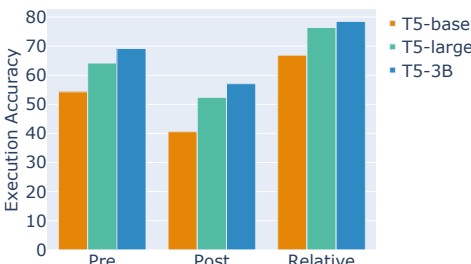 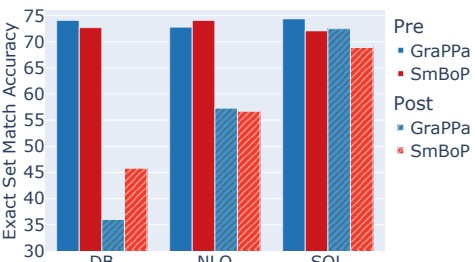

Figure 2: Pre-perturbation, post-perturbation, and relative robustness accuracy of T5-BASE, T5-LARGE, and T5-3B in terms of EX.

Figure 3: EM accuracy of GRAPPA and SMBOP on pre-perturbation and post-perturbation data of DB, NLQ, and SQL.

models have both a higher post-perturbation accuracy and a relative robustness accuracy, which indicates that using large pre-trained models is one solution for improving the robustness of models.

**What are the advantages of top-down and bottom-up decoders?**  Figure 3 shows the robustness of GRAPPA and SMBOP which have the same encoder but with a top-down decoder and a bottom-up decoder. We find that the bottom-up decoder model SMBOP is more robust to DB perturbations, and the top-down decoder model GRAPPA is more robust to NLQ perturbations. We believe it is because the DB perturbation modifies local inputs of the database schema and the bottom-up decoder decodes the leaf nodes first, which learns a better alignment between NLQ and databases. However, NLQ perturbations are via sentence-level paraphrasing, where the top-down decoder decodes the structure of SQL first, which is more robust than sentence-level paraphrases. This implies that a model combining the top-down and bottom-up decoders could be a solution to improve models' robustness on text-to-SQL tasks.

**Does question and DB content linking benefit model robustness?**  The SOTA models leverage an entity linking feature between question tokens and DB content based on string matching to improve value prediction performance, which concatenates column names with mentioned DB contents in input (Lin et al., 2020; Scholak et al., 2021). This feature provides two benefits: (1) indicating which column should be predicted in SQL by its mentioned DB content and (2) providing the format of a mentioned value in its DB. We compare T5-3B and T5-3B LK against NLQ perturbations and report their relative robustness accuracy. Using the entity linking significantly improves model robustness against column-value (68.7 vs 78.1) and value-synonym (35.8 vs 46.1) perturbations in terms of EX accuracy. The entity linking benefits the SQL value prediction by giving how the value is presented in the DB especially when the value appears in a different format in NLQ (value-synonym). However, this feature slightly hurts the SQL prediction in terms of EM accuracy (ignoring values) on value-synonym (66.2 vs 67.2). We believe that it is because the model overfits the string-matching feature to indicate the mentioned column and does not generalize well when the matching fails due to perturbation. Therefore, it is beneficial to provide the formats of mentioned values in the DB via the linking. And a better linking mechanism should be adapted for developing robust text-to-SQL models. A detailed analysis can be found in Appendix B.

## 6  CONCLUSION AND FUTURE WORK

We curate a diagnostic text-to-SQL evaluation benchmark Dr.Spider, which contains various perturbations on DB, NLQ, and SQL to simulate diverse task-specific robustness challenges. We diagnose the robustness of SOTA text-to-SQL models regarding their fine-gained components with our benchmark. Our experiments reveal existing models are vulnerable to perturbations, while some model architecture designs are more robust against certain perturbations. We envision our findings could benefit future model design for robust text-to-SQL models. In the future, we want to explore different strategies of prompt examples selection besides handpicking for paraphrasing NLQ. We also plan to improve the robustness of text-to-SQL models with the insight from this work, such as combining top-down and bottom-up decoder and using our perturbation methods for data augmentation.

## 7 Ethics Statement

We acknowledge the importance of the ICLR Code of Ethics and agree with it. An unrobust text-to-SQL system may cause harm by misinforming its users. Our work aims at addressing it by providing a diagnostic robustness evaluation platform and insights to improve the robustness of models. We construct our benchmark Dr.Spider based on Spider, an public and free text-to-SQL benchmark. We recruited annotators from Amazon Mechanical Turk (MTurk) for the crowdsourcing paraphrase study as well as the clarity and fluency annotation for our data quality control. We took great care to pay fair wages. Crowdsourcing annotators are required to paraphrase a question or label the clarity and fluency of a question. No sensitive information about annotators was asked and the only personal information we collected was the worker IDs on Mturk, which will not be released. We leverage large pre-trained language models (PLM) to paraphrase questions as our NLQ perturbations. We acknowledge that PLM may generate toxic language (Sheng et al., 2019; Gehman et al., 2020). Three expert annotators carefully review all generated paraphrases to ensure no offensive language in our dataset.

## 8 Acknowledgement

We would like to thank Eric Fosler-Lussier, Michael White, Micha Elsner, Srinivasan Parthasarathy, Huan Sun, Yu Su, Amad Hussain, Pengfei Liu, and others at the OSU SLaTe lab and the Clippers seminar, as well as the anonymous reviewers for their valuable feedback on this work.

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

## A DATA DETAILS

### A.1 CROWDSOURCING PARAPHRASE STUDY INTERFACE

To understand the linguistic phenomena that NLQs could have in a realistic scenario, we recruit annotators on Amazon Mechanical Turk to paraphrase 100 sampled questions in Spider and collect five paraphrases per question. Figure 4, 5 contain the instruction and interface on Amazon Mechanical Turk for crowdsourcing paraphrase collection. Annotators are required to rephrase a question in their own words which do not change the meaning of the question given the context. We provide a short description of the corresponding database as the context of a question.

**Paraphrase Question**

In this task, you will be given a English question and its context. Please paraphrase the question in your own words. The answer to the paraphrased question should be the same as the given question (do not omit information in the original question).

Below are 2 good paraphrase examples:
**Context:** There is a database containing airlines, flights and airports.
**Question:** What is the code of airport that has the highest number of flights?

**Good Paraphrase 1:** Which airport has the most flights? List its code.
**Good Paraphrase 2:** What is the airport code with the largest amount of flights?

**Question:** What is the abbreviation of the airilne has the fewest flights and what country is it in?

**Good Paraphrase 1:** What is the code of airline that has the least number of flights and where does this airline operate?
**Good Paraphrase 2:** Show me the abbreviation code and country of the airline that has the least flights.

Below are a unaccepted example:
**Question:** What is the abbreviation of the airilne has the fewest flights and what country is it in?

**Bad Paraphrase 1:** What is the abbreviation of the airilne what country is it in?
(It is not a good paraphrase because the criteria **fewest flights** in the original question is omitted)
**Bad Paraphrase 2:** What is the details of the airilne with the fewest amount of flights?
(It is not a good paraphrase because **details** does not mean the same as **abbreviation of the airilne** and **country** in the original question is omitted)

Figure 4: The instructions for crowdsourcing paraphrase collection on Amazon Mechanical Turk.

Paraphrase the question **without changing its original meaning** in the given context: (Assume the answer to the paraphrased question should be the same as the given question)

**Context:** A database containing WTA players information, their ranking and match records.

**Question:** find the names of loser and winner who played in the match with greatest number of minutes.

Paraphrase:

Type your paraphrase here. Thank you.

**Submit**

Figure 5: The interface for crowdsourcing paraphrase collection on Amazon Mechanical Turk. Annotators are given a free text box to input their paraphrased question.

### A.2 PARAPHRASE GENERATION DETAILS

Each NLQ paraphrase category contains an individual prompt prefix which comprises 5 paraphrase examples and a question to be paraphrased, except for `Multitype` which includes examples that involve multiple phenomena selected from other categories. Each paraphrase example contains 4 parts: (1) an original sentence, (2) the mentioned SQL tokens that need to be rephrased or implied, (3) a paraphrased question, and (4) an explanation of how the question is paraphrased. We use OPT-66B to generate paraphrases given an original question. Inspired by Wei et al. (2022); Paranjape et al. (2021), we employ the OPT model to generate an explanation along with the paraphrase given an original question and the mentioned SQL tokens related to the category. Table 4 is the prompt prefix of the perturbation `column-value`. The prompt prefixes for other categories are available at `https://github.com/awslabs/diagnostic-robustness-text-to-sql`.

As Bandel et al. (2022) discovered, there is a trade-off between paraphrase diversity and factuality. We use multiple combinations of hyper-parameters in paraphrase generation to take both into account. For each question, we run the OPT model 4 times with the hyperparameters top_p in {0.9,1.0}

and temperature in {0.7,1.0}, and 5 paraphrases are returned each time (Keskar et al., 2019). We implement the paraphrase generation with Huggingface Wolf et al. (2019). The experiments were done on 8 Nvidia Tesla V100 with 32G memory for about 5 days.

---

Paraphrase the sentence by implicitly mentioning the type with its entity/number:

(0) **Sentence**: Show the stadium name and capacity with most number of concerts in year 2014 or after. **Mentioned type and entity**: 2014 is a year. **Paraphrase**: What is the name and capacity of the stadium which held the most concerts in 2014 or later? **Explanation**: remove year in paraphrase.

(1) **Sentence**: What are the ids of the students who do not own cats as pets? **Mentioned type and entity**: cat is a pet type. **Paraphrase**: Find the IDs of students who don't own cats. **Explanation**: remove as pets in paraphrase.

(2) **Sentence**: How many 'United Airlines' flights depart from Airport 'AHD'? **Mentioned type and entity**: United Airlines is a airline name; AHD is a source airport. **Paraphrase**: Of the flights out of 'AHD', how many were 'United Airlines'? **Explanation**: remove Airport in paraphrase.

(3) **Sentence**: What are the paragraph texts for the document with the name 'Customer reviews'? **Mentioned type and entity**: Customer reviews is a document name. **Paraphrase**: What is the paragraph text in the document 'Customer reviews'? **Explanation**: remove name in paraphrase.

(4) **Sentence**: What is the average expected life expectancy for countries in the region of Central Africa? **Mentioned type and entity**: Central Africa is a region. **Paraphrase**: For countries in central Africa what is the average life expectancy? **Explanation**: remove region in paraphrase.

(5) **Sentence**: [new sentence] **Mentioned type and entity**: [mentioned DB content in SQL] is a [corresponding column]. **Paraphrase**:

---

Table 4: Prompt prefix for `column-value`.

### A.3 PARAPHRASE FILTERING AND QUALITY CONTROL DETAILS

We use a simple and high-recall rule-based classifier to remove out-of-category paraphrases based on the alignment between SQL tokens and their indicators in NLQ. In particular, we use the annotations from Lei et al. (2020) to build the column and value indicators. And to obtain SQL keyword indicators, we collect the common indicators (covering over 80% examples) of each SQL keyword in the NLQs and programmatically label NLQs with those indicators. Table 5 contains the common indicators of SQL keywords in Spider. We filter out the paraphrases where all the indicators for SQL keywords, columns and values remain the same as the original NLQ except category `others`.

To ensure the data quality in Dr.Spider, we request three annotators with SQL expertise to carefully review paraphrases with gold SQL labels and select the final data. Table 6 contains the randomly sampled NLQ paraphrases in each category and the annotator's decision.

To measure the naturalness of our generated NLQ perturbations, we sample 500 questions from each of the original Spider development set, human paraphrases (we use all 270 factual paraphrases), and our perturbed questions. We hire crowdsourcing annotators on Amazon Mechanical Turk to score the clarity and fluency of sampled questions from 1 to 3. Clarity=1 represents that the annotator couldn't understand the question, clarity=2 indicates that the question is ambiguous to the annotator, and clarity=3 means that the question is clear. Fluency=1 represents that the question is full of grammar errors or typos, fluency=2 indicates the question has minor errors, and fluency=3 means the question is fluent without grammar errors or typos. Figure 6 is the interface for labeling the clarity and fluency of questions in the original Spider dataset, collected human paraphrases, and the questions in Dr. Spider.

| SQL Keyword | Indicator |
|---|---|
| > | more than, larger than, bigger than, higher than, above, after, older than, heavier than |
| < | less than, smaller than, lower than, below, before, younger than, lighter than |
| >= | at most, or more |
| <= | at least, or less |
| between and | between |
| count() | number, amount, count |
| sum() | total, amount |
| avg() | average, mean |
| max() | maximum, most, highest |
| min() | minimum, least, lowest |
| ASC | ascending, in alphabetical order, in lexicographical order, from youngest to oldest, from young to old, from low to high |
| ASC LIMIT | least, lowest, smallest, youngest, earliest, shortest, minimum, fewest number, fewest amount |
| DESC | descending, in reverse alphabetical order, in reversed lexicographical order, from the oldest to the youngest, from old to young, from high to low |
| DESC LIMIT | most, highest, largest, oldest, latest, longest, maximum, greatest number, greatest amount |

Table 5: The common NLQ indicator tokens for SQL keywords.

## Determine the clarity and fluency of a question under context

**Guide:** Read the context section first. The questions below is related to the tables in the context. Read the question carefully and determine the clarity and fluency of the question.

**Context:**

A database containing the information of all countries in the world as well as major cities of each country.

**Table city**

| id | name | countrycode | district | population |
|----|------|-------------|----------|-----------|
| 1 | Kabul | AFG | Kabol | 1780000 |
| 2 | Qandahar | AFG | Qandahar | 237500 |
| 3 | Herat | AFG | Herat | 186800 |
| ... | ... | ... | ... | ... |

**Table country**

| code | name | continent | region | surfacearea | indepyear | population | lifeexpectancy | gnp | gnpold | localname | governmentform | headofstate | capital | code2 |
|------|------|-----------|--------|-------------|-----------|------------|----------------|-----|--------|-----------|----------------|-------------|---------|-------|
| ABW | Aruba | North America | Caribbean | 193.0 | None | 103000 | 78.4 | 828.0 | 793.0 | Aruba | Nonmetropolitan Territory of The Netherlands | Beatrix | 129 | AW |
| AFG | Afghanistan | Asia | Southern and Central Asia | 652090.0 | 1919 | 22720000 | 45.9 | 5976.0 | None | Afganistan/Afqanestan | Islamic Emirate | Mohammad Omar | 1 | AF |
| AGO | Angola | Africa | Central Africa | 1246700.0 | 1975 | 12878000 | 38.3 | 6648.0 | 7984.0 | Angola | Republic | José Eduardo dos Santos | 56 | AO |
| ... | ... | ... | ... | ... | ... | ... | ... | ... | ... | ... | ... | ... | ... | ... |

**Table countrylanguage**

| countrycode | language | isofficial | percentage |
|-------------|----------|------------|------------|
| ABW | Dutch | T | 5.3 |
| ABW | English | F | 9.5 |
| ABW | Papiamento | F | 76.7 |
| ... | ... | ... | ... |

---

**Question:**

Find the number of cities in each district whose population is greater than the average population of cities?

**Determine the clarity of the above question**
○ I don't know what this question is asking about.
○ The question is mostly clear, but there are still some ambiguities. (There is more than one way to answer this question.)
○ The question is clear and NOT ambiguous.

**Determine the fluency of the above question**
○ The question is full of grammar errors or typos in English, or it has an imcomplete sentence.
○ The question is fluent overall, but has one or two minor grammar errors or typos such as plural form, spelling, tenses etc.
○ The question is fluent and free from grammar errors nor typos.

Submit

Figure 6: Annotation interface for labeling the clarity and fluency for a given question.

| Paraphrase Example |
| --- |
| `Keyword-synonym`
Question: find the code of the country where has the greatest number of players.
Paraphrase: Display the code of the country with the largest player population.
*Selected in Dr.Spider* |
| `Keyword-carrier`
Question: Count the number of dogs that went through a treatment.
Paraphrase: How many dogs have been treated?
*Selected in Dr.Spider* |
| `Column-synonym`
Question: What is the average weight for each type of pet?
Paraphrase: For each category of pets, what is the average weight?
*Selected in Dr.Spider* |
| `Column-synonym`
Question: What is the money rank of the poker player with the highest earnings?
Paraphrase: The poker player at rank n made the most income during his or her career.
*Not Selected in Dr.Spider* |
| `Column-carrier`
Question: Find the name of tourney that has more than 10 matches.
Paraphrase:Which tournament has more than 10 games?
*Selected in Dr.Spider* |
| `Column-carrier`
Question: What are the song titles and singer names?
Paraphrase: Which songs have titles and singers?
*Not selected in Dr.Spider* |
| `Column-attribute`
Question: What is the car model with the highest mpg ?
Paraphrase: what is the car model that is most gas efficient?
*Selected in Dr.Spider* |
| `Column-value`
Question: What is the phone number of the man with the first name Timmothy and the last name Ward?
Paraphrase: Find the phone number for Timmothy Ward?
*Selected in Dr.Spider* |
| `Value-synonym`
Question:What are the names of conductors whose nationalities are not "USA"?
Paraphrase: Names of conductors whose nationalities are not US.
*Selected in Dr.Spider* |
| `Multitype`
Question: What are names of countries with the top 3 largest population?
Paraphrase: Which three countries have the highest population?
*Selected in Dr.Spider* |
| `Multitype`
Question: What are the airline names and abbreviations for airlines in the USA?
Paraphrase: list all the airlines and identify their respective codes in the United States.
*Not selected in Dr.Spider* |
| `Others`
Question: What is the average, minimum, and maximum age of all singers from France?
Paraphrase: Show me your data on the average, minimum, and maximum ages of all singers from France.
*Selected in Dr.Spider* |

Table 6: Randomly sampled paraphrases and human annotator selection decision.

## A.4 SQL PERTURBATION

| SQL Perturbation | Pre-perturbation | Post-perturbation |
|---|---|---|
| Comparison | SQL: `HAVING count(*) > 1`
NLQ: more than one car maker? | SQL: `HAVING count(*) >= 1`
NLQ: at least one car maker? |
| Sort-order | SQL: `ORDER BY age DESC`
NLQ: from the oldest to the youngest. | SQL: `ORDER BY age ASC`
NLQ: from the youngest to the oldest. |
| NonDB-number | SQL: `ORDER BY winner_age LIMIT 3`
NLQ: 3 youngest winners | SQL: `ORDER BY winner_age LIMIT 5`
NLQ: 5 youngest winners |
| DB-text | SQL: `country = 'France'`
NLQ: singers from France | SQL: `country = 'Netherlands'`
NLQ: singers from Netherlands |
| DB-number | SQL: `horsepower > 150`
NLQ: horsepower more than 150 | SQL: `horsepower > 145`
NLQ: horsepower more than 145 |

Table 7: SQL perturbations and examples.

Table 7 shows an example of each SQL perturbation. Since SQL perturbations involve modifications on both SQL tokens and their indicators in NLQs, we want to minimize the surface-level changes introduced to NLQs to disentangle SQL perturbations from NLQ perturbations. For `comparison` and `sort-order` perturbations, we programmatically modify the keywords in SQL and their indicators in NLQ following the rules in Tables 8 - 10. We replace only one indicator with another that contains the same meaning in the context. For example, when replacing > with <, "older than" is replaced by "younger than" instead of "lighter than".

| > | < | >= | <= |
|---|---|---|---|
| above | below | - | - |
| after | before | - | - |
| older than | younger than | - | - |
| heavier than | lighter than | - | - |
| more than | less than | at most | at least |
| larger/bigger than | smaller than | - | - |
| higher than | lower than | - | - |
| - | - | or more | or less |

Table 8: The common NLQ indicator tokens for comparison operations in Spider.

| ORDER BY ... ASC | ORDER BY ... DESC |
|---|---|
| ascending | descending |
| in alphabetical order | in reverse alphabetical order |
| in lexicographical order | in reversed lexicographical order |
| from the youngest to the oldest | from the oldest to the youngest |
| from young to old | from old to young |
| from low to high | from high to low |

Table 9: The common NLQ indicator tokens for ascending sort and desending sort in Spider.

| ORDER BY ... ASC LIMIT | ORDER BY ... DESC LIMIT |
|:---:|:---:|
| least | most |
| lowest | highest |
| smallest | largest |
| youngest | oldest |
| earliest | latest |
| shortest | longest |
| minimum | maximum |
| fewest number | greatest number |
| fewest amount | greatest amount |

Table 10: The common NLQ indicator tokens for ascending sort and descending sort followed by a `LIMIT` clause in Spider.

## B  MORE EVALUATION RESULTS

Table 13 shows the EM accuracy of SOTA models on pre-perturbation and post-perturbation data in Dr.Spider. EM accuracy does not evaluate value predictions in SQLs. Therefore, EM is not effective to measure model robustness against `NonDB-number`, `DB-text`, and `DB-text`. Table 14, 15 contains the relative robustness accuracy on EX and EM of SOTA text-to-SQL models. Relative robustness accuracy ignores the examples that the model predicts wrong before perturbations for each model.

Table 11 contains the subcategories and examples in `DBcontent-equivalence`. `DBcontent-equivalence` simulate a scenario where a column content is represented in different formats by replacing a column with one or multiple semantic-equivalent columns. Table 12 contains prediction examples of PICARD on `DBcontent-equivalence` perturbation data. PICARD predicts SQLs correctly for all four examples before perturbations. The first two examples both replace column `age` with `birthyear`, however, PICARD defenses against one perturbation example but fails on the other. The third example replaces a fix-value text column `pettype` with two boolean columns `is_dog` and `is_cat`. PICARD fails to predict the right value for the boolean columns `is_dog`. The last example replaces a compound column `name` with two columns `firstname` and `lastname`. PICARD refer both `firstname` and `lastname` when the NLQ asks for "names". We believe that `DBcontent-equivalence` requires models to have additional common-sense reasoning to understand the relation between a database column and how it is represented in NLQ. This phenomenon is common in reality when the actual users are not aware of DB schemas.

| Category | Column Example |
|---|---|
| Single text to multiple texts | `name => firstname, lastname` |
| Boolean to text | `is_male => gender` |
| Text to boolean | `hand => is_right_hand` |
| Single text to multiple booleans | `degree_summary => is_bachelor, is_master, is_PhD` |
| Number to number | `age => birthyear` |

Table 11: Categories and examples in the `DBcontent-equivalence` perturbation.

**The effectiveness of entity linking between NLQ and DB content**  Figure 7 shows the relative robustness accuracy on EX and EM of T5-3B models with and without the entity linking feature. Using the entity linking significantly improves model robustness against `column-value` and `value-synonym` perturbations in terms of EX while it gives less robustness on `column-attribute` and `column-carrier` perturbations (both EX and EM).

The EX and EM accuracy on the `value-synonym` perturbation are different. The entity linking benefits the SQL value prediction by giving how the value is presented in the DB especially when the value appears in a different format in NLQ (value-synonym). However, this feature slightly hurts the SQL prediction in terms of EM accuracy (ignoring values) on value-synonym (66.2 vs 67.2). We believe that it is because the model overfits the string-matching feature to indicate the mentioned column and does not generalize well when the matching fails due to perturbation. Therefore, it is beneficial to provide the formats of mentioned values in the DB via the linking. And a better linking mechanism should be adapted for developing robust text-to-SQL models

**Robustness Evaluation on in-context learning models**  Besides the models that are trained on the Spider dataset, we also evaluate the robustness of a large language model, CODEX Chen et al. (2021), for in-context learning. We use CODEX DAVINCI which finetunes GPT-3 Brown et al. (2020) on GitHub codes. We choose the `Create Table + Select 3` prompt from (Rajkumar et al., 2022), as it achieves the best performance with in-context learning (74.0 execution accuracy on the original Spider development set based on our implementation), which is even comparable with the SOTA finetuned model, PICARD (79.3 execution accuracy). Table 16 shows the comparison between PICARD and CODEX on their pre-perturbation, post-perturbation accuracy, and relative robustness accuracy. We have a few findings: (1) In-context CODEX is more robust than finetuned PICARD

| Example |
| --- |
| **NLQ**: *What are the names and release years for all the songs of the youngest singer?* 
 **Gold SQL**: `...ORDER BY age LIMIT 1` 
 **Column Substitution**: `age -> birthyear` 
 **Gold SQL after perturbation**: `...ORDER BY birthyear DESC LIMIT 1` 
 **Predicted SQL after perturbation**: `...ORDER BY age LIMIT 1` 
 *Incorrect* |
| **NLQ**: *Find the type and weight of the youngest pet.* 
 **Gold SQL**: `...ORDER BY pet_age LIMIT 1` 
 **Column Substitution**: `age -> birthyear` 
 **Gold SQL after perturbation**: `...ORDER BY pet_birthyear DESC LIMIT 1` 
 **Predicted SQL after perturbation**: `...ORDER BY pet_birthyear DESC LIMIT 1` 
 *Correct* |
| **NLQ**: *How many dog pets are raised by female students?* 
 **Gold SQL**: `...WHERE Pets.pettype = 'dog'...` 
 **Column Substitution**: `pettype -> is_dog` 
 **Gold SQL after perturbation**: `...WHERE Pets.is_dog = True...` 
 **Predicted SQL after perturbation**: `...WHERE Pets.is_dog = 'D'...` 
 *Incorrect* |
| **NLQ**: *List all singer names in concerts in year 2014.* 
 **Gold SQL**: `SELECT name FROM...` 
 **Column Substitution**: `name -> first_name, last_name` 
 **Gold SQL after perturbation**: `SELECT first_name, last_name FROM...` 
 **Predicted SQL after perturbation**: `SELECT first_name, last_name FROM...` 
 *Correct* |

Table 12: Prediction examples of PICARD on `column-equivalence` perturbation data.

to database perturbations even though it still suffers a performance drop from 72.6 to 60.7, and (2) in-context CODEX does not show better robustness against NLQ perturbations than finetuned PICARD.

We hypothesize that the robustness of CODEX is largely related to its training data. As the training data are obtained from GitHub codes, CODEX has seen a diverse distribution of databases. Specifically, we find that CODEX has a limited performance drop to `Schema-abbreviation` perturbation (from 70.2 to 68.6), compared to PICARD (from 74.9 to 64.7). We believe that CODEX is robust to the `Schema-abbreviation` perturbation, as it has seen a large number of abbreviation schemas in its training data.

In NLQ perturbations, we find two categories that are particularly challenging to CODEX: `Column-carrier` (from 80.8 to 51.1), and `Column-attribute` (from 68.9 to 46.2). Those perturbations implicitly mention a column with the question carrier or the attribute of the column. For example, In Table 1, the column "name" in "Show the name of teacher" is implied by "Which teachers"" and the column "year" in "worked the greatest number of year" is implied by "worked the longest". CODEX was trained with the goal of generating code from docstrings. The language style of common docstrings is slightly different from the style of spoken language therefore CodeX may not be familiar with such implicit mentions. As the CodeX details are not fully published, the analysis is subjective.

| | Perturbation | RatSQL | GraPPa | SmBop | T5-base | T5-large | T5-3B | T5-3B LK | Picard |
|---|---|---|---|---|---|---|---|---|---|
| | Spider-dev | 69.5* | 73.4 | 74.7 | 56.0* | 65.3 | 68.8* | 71.5 | 75.5 |
| DB | Schema-synonym | 63.2—41.1 | 67.1—49.4 | 67.5—49.4 | 48.5—24.3 | 57.7—36.0 | 61.9—40.7 | 63.7—45.2 | **68.7**—**54.1** |
| | Schema-abbreviation | 65.2—40.0 | 69.7—53.3 | 69.3—54.1 | 49.2—23.1 | 62.0—38.3 | 64.3—49.9 | 67.6—50.7 | **72.7**—**60.4** |
| | DBcontent-equivalence | 81.2—5.8 | **85.6**—5.2 | 81.2—34.0 | 58.4—15.2 | 71.7—30.1 | 77.0—34.8 | 82.5—36.4 | 85.3—**39.8** |
| | Avergae | 69.9—29.0 | 74.1—36.0 | 72.7—45.8 | 52.0—20.9 | 63.8—34.8 | 67.7—41.8 | 71.3—44.1 | **75.6**—**51.4** |
| NLQ | Keyword-synonym | 63.4—49.3 | 70.4—55.5 | **73.0**—58.1 | 51.4—41.2 | 63.6—52.5 | 64.2—52.6 | 67.9—55.6 | 70.2—**58.7** |
| | Keyword-carrier | **79.2**—**76.2** | 78.9—74.2 | 74.9—72.7 | 62.9—59.4 | 70.4—66.2 | 72.9—66.7 | 76.9—71.9 | 77.9—75.9 |
| | Column-synonym | 67.9—47.8 | 71.2—54.4 | 67.7—50.1 | 48.7—32.7 | 55.4—44.4 | 66.6—50.3 | 67.9—54.5 | **74.6**—**59.7** |
| | Column-carrier | 74.8—55.3 | 81.0—**61.5** | 78.6—57.2 | 57.0—39.4 | 71.2—56.1 | 73.6—55.3 | 79.6—54.4 | **83.6**—57.0 |
| | Column-attribute | 51.3—34.5 | 58.0—48.7 | **72.3**—**53.8** | 42.9—37.8 | 49.6—38.7 | 54.6—46.2 | 55.5—45.4 | 64.7—52.9 |
| | Column-value | 78.0—55.3 | 82.6—51.6 | **84.2**—**63.8** | 52.3—28.3 | 68.4—44.1 | 68.8—43.1 | 65.8—52.6 | 78.3—62.5 |
| | value-synonym | 60.3—38.1 | 66.8—53.6 | 67.4—40.7 | 43.7—33.6 | 56.7—46.4 | 62.1—43.7 | 67.8—47.0 | **70.4**—**58.3** |
| | Multitype | 67.1—42.0 | 72.8—51.5 | **74.2**—46.0 | 50.1—33.5 | 61.8—44.9 | 66.1—44.9 | 69.9—48.9 | 73.2—**55.4** |
| | Others | 71.5—63.5 | 73.1—65.1 | 74.7—67.6 | 56.9—49.7 | 65.4—60.0 | 68.4—62.2 | 72.1—66.1 | **75.5**—**70.5** |
| | Average | 68.2—51.3 | 72.8—57.3 | 74.1—56.7 | 51.8—39.5 | 62.5—50.4 | 66.4—51.7 | 69.3—55.2 | **74.3**—**61.2** |
| SQL | Comparison | 56.2—53.9 | **68.0**—**64.0** | 63.5—57.9 | 46.1—28.7 | 54.5—55.1 | 58.4—51.7 | 59.0—55.6 | 61.2—58.4 |
| | Sort-order | 64.6—65.6 | **75.5**—**76.0** | 73.4—73.4 | 58.9—58.9 | 68.8—66.1 | 70.8—69.3 | 71.9—67.7 | **75.5**—72.4 |
| | NonDB-number | 66.4—67.2 | 75.6—72.5 | 73.3—68.7 | 58.0—56.5 | 62.6—65.6 | 77.9—**77.9** | 79.4—74.0 | **81.7**—74.8 |
| | DB-text | 63.6—65.0 | **67.0**—64.2 | 66.8—62.2 | 43.5—40.6 | 54.2—53.1 | 58.3—54.1 | 59.3—55.4 | 66.3—**65.5** |
| | DB-number | 72.9—72.9 | **86.1**—**85.9** | 83.7—82.2 | 66.8—64.4 | 75.4—74.9 | 78.8—79.8 | 83.9—83.4 | 83.9—83.4 |
| | Average | 64.7—64.9 | **74.4**—**72.5** | 72.1—68.9 | 54.7—49.8 | 63.1—63.0 | 68.8—66.6 | 70.7—67.2 | 73.7—70.9 |
| All | - | 67.5—51.4 | 73.5—58.0 | 73.3—58.3 | 52.7—39.3 | 62.9—51.3 | 67.3—54.3 | 70.0—56.8 | **74.3**—**62.3** |

Table 13: The exact set match (EM) accuracy of SOTA text-to-SQL models on the original Spider development set (Spider-dev) and Dr.Spider. * represents the model we trained with official codes as their models are not public. x—y represents the pre-perturbation accuracy and post-perturbation accuracy (i.e. absolute robustness accuracy ). Bold number is most highest absolute robustness accuracy in each category and underscores stand for the second best results. Note that the EM accuracy does not evaluate value predictions. Therefore, it is not effective to measure model robustness against `NonDB-number`, `DB-text`, and `DB-text`.

| | Perturbation | RatSQL | SmBop | T5-base | T5-large | T5-3B | T5-3B LK | Picard |
|---|---|---|---|---|---|---|---|---|
| DB | Schema-synonym | 64.6 | 73.5 | 46.1 | 58.5 | 62.8 | 67.2 | **74.6** |
| | Schema-abbreviation | 62.2 | 79.3 | 41.4 | 59.4 | 72.0 | 72.7 | **83.0** |
| | DBcontent-equivalence | 11.8 | 42.6 | 27.1 | 44.0 | 44.1 | 46.1 | **49.3** |
| | Avergae | 46.2 | 65.1 | 38.2 | 54.0 | 59.6 | 62.0 | **69.0** |
| NLQ | Keyword-synonym | 73.9 | 81.5 | 78.8 | 81.9 | 84.8 | 85.7 | **88.0** |
| | Keyword-carrier | 94.6 | 93.0 | 90.9 | 92.6 | 93.0 | 92.4 | **96.2** |
| | Column-synonym | 62.4 | 68.5 | 56.9 | 65.8 | 67.0 | 73.3 | **77.7** |
| | Column-carrier | 67.7 | 73.2 | 67.2 | 74.2 | **74.5** | 71.9 | 73.0 |
| | Column-attribute | 65.6 | 70.0 | 67.7 | 80.7 | **89.1** | 83.1 | 82.9 |
| | Column-value | 70.1 | 72.3 | 50.6 | 70.8 | 68.7 | 78.1 | **81.7** |
| | value-synonym | 25.4 | 37.5 | 49.1 | 55.2 | 52.5 | 63.7 | **69.2** |
| | Multitype | 52.5 | 58.6 | 59.4 | 64.3 | 65.5 | 69.1 | **72.8** |
| | Others | 84.6 | 89.3 | 82.9 | 87.5 | 88.7 | 94.0 | **96.1** |
| | Average | 66.3 | 71.5 | 67.1 | 74.8 | 76.0 | 79.0 | **82.0** |
| SQL | Comparison | 82.1 | 91.0 | 60.7 | 91.2 | 90.5 | **92.9** | 89.3 |
| | Sort-order | 88.6 | 93.1 | 91.6 | 89.5 | **95.8** | 91.7 | 92.1 |
| | NonDB-number | 73.5 | 91.1 | 89.7 | 93.9 | **100.0** | 94.1 | 92.7 |
| | DB-text | 79.2 | 91.1 | 83.3 | 90.0 | 88.6 | 90.8 | **94.1** |
| | DB-number | 98.7 | 95.5 | 93.9 | 98.7 | 97.6 | **99.7** | 98.3 |
| | Average | 84.4 | 92.4 | 83.8 | 92.7 | **94.5** | 93.8 | 93.3 |
| All | - | 68.1 | 76.5 | 66.9 | 76.4 | 78.5 | 80.4 | **83.0** |

Table 14: The relative robustness accuracy on EX of SOTA text-to-SQL models on Dr.Spider. Bold number is most highest absolute robustness accuracy in each category and underscores stand for the second best results.

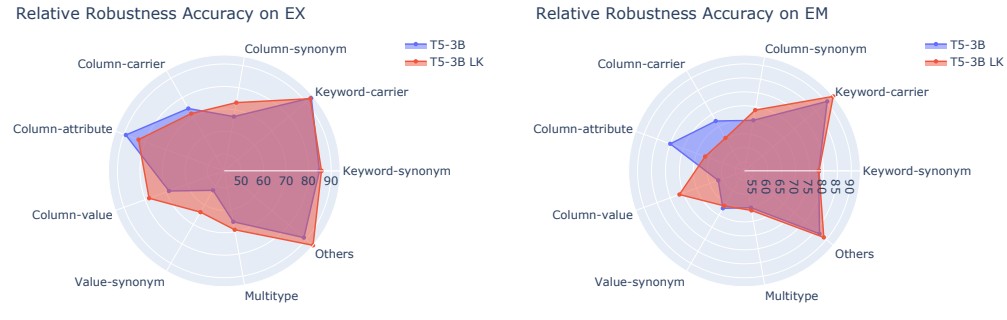

(a) Relative Robustness Accuracy on EX   (b) Relative Robustness Accuracy on EM

Figure 7: Relative robustness accuracy on EX and EM of T5-3B models with and without question and DB content linking.

| | Perturbation | RatSQL | GraPPa | SmBop | T5-base | T5-large | T5-3B | T5-3B LK | Picard |
|---|---|---|---|---|---|---|---|---|---|
| DB | Schema-synonym | 61.1 | 70.3 | 71.0 | 46.0 | 59.3 | 63.7 | 67.8 | **75.9** |
| | Schema-abbreviation | 58.9 | 73.4 | 76.2 | 38.6 | 59.3 | 74.0 | 72.2 | **80.8** |
| | DBcontent-equivalence | 6.5 | 5.5 | 38.1 | 22.4 | 38.0 | 41.5 | 43.5 | **46.6** |
| | Avergae | 42.2 | 49.7 | 61.8 | 35.7 | 52.2 | 59.7 | 61.2 | **67.8** |
| NLQ | Keyword-synonym | 72.2 | 75.3 | 77.2 | 73.5 | 77.6 | 78.3 | 78.4 | **80.0** |
| | Keyword-carrier | 93.0 | 93.0 | 94.0 | 91.2 | 91.8 | 90.4 | 93.2 | **95.8** |
| | Column-synonym | 66.5 | 73.1 | 69.8 | 62.0 | 71.8 | 70.1 | 73.8 | **77.6** |
| | Column-carrier | 67.0 | **74.2** | 68.6 | 65.5 | 73.8 | 72.3 | 65.3 | 66.3 |
| | Column-attribute | 65.6 | 79.7 | 73.3 | 78.4 | 67.8 | **80.0** | 66.7 | 67.5 |
| | Column-value | 66.7 | 60.2 | 75.8 | 54.1 | 63.9 | 61.7 | 76.5 | **78.6** |
| | value-synonym | 57.7 | 79.0 | 57.8 | 68.3 | 73.2 | 67.2 | 66.2 | **79.2** |
| | Multitype | 59.4 | 68.9 | 60.5 | 62.8 | 67.1 | 65.1 | 66.1 | **72.1** |
| | Others | 83.5 | 85.2 | 86.8 | 81.6 | 86.3 | 86.8 | 88.8 | **90.9** |
| | Average | 70.2 | 76.5 | 73.8 | 70.8 | 74.8 | 74.7 | 75.0 | **78.7** |
| SQL | Comparison | 86.0 | 88.4 | 87.6 | 58.5 | **92.8** | 85.6 | 91.4 | 89.9 |
| | Sort-order | 93.5 | **97.2** | 92.9 | 93.8 | 93.9 | 97.1 | 92.8 | 93.8 |
| | NonDB-number | 96.6 | 96.0 | 91.7 | 94.7 | **100.0** | **100.0** | 92.3 | 91.6 |
| | DB-text | 94.5 | 92.3 | 90.1 | 86.9 | 94.1 | 91.7 | 90.2 | **95.5** |
| | DB-number | 99.3 | 98.6 | 98.3 | 95.3 | **99.4** | 98.5 | 99.1 | 99.1 |
| | Average | 94.0 | 94.5 | 92.1 | 85.8 | **96.0** | 94.6 | 93.2 | 94.0 |
| All | - | 72.2 | 77.1 | 77.0 | 69.0 | 77.1 | 77.9 | 77.9 | **81.2** |

Table 15: The relative robustness accuracy on EM of SOTA text-to-SQL models on Dr.Spider. Bold number is most highest absolute robustness accuracy in each category and underscores stand for the second best results. Note that the EM accuracy does not evaluate value predictions. Therefore, it is not effective to measure model robustness against NonDB-number, DB-text, and DB-text.

| | Perturbation | Pre-pertubation—Post-perturbation | | Relative Robustness | |
| --- | --- | --- | --- | --- | --- |
| | | **PICARD** | **CODEX** | **PICARD** | **CODEX** |
| | Spider-dev | **79.3** | 74.0 | - | - |
| DB | Schema-synonym | **73.0**—56.5 | 68.9—**62.0** | 74.6 | **84.6** |
| | Schema-abbreviation | **74.9**—64.7 | 70.2—**68.6** | 83.0 | **92.8** |
| | DBcontent-equivalence | **88.7**—43.7 | 78.8—**51.6** | 49.3 | **62.5** |
| | Average | **78.9**—55.0 | 72.6—**60.7** | 69.0 | **80.0** |
| NLQ | Keyword-synonym | **72.6**—**66.3** | 63.8—55.5 | **88.0** | 81.7 |
| | Keyword-carrier | 85.0—82.7 | **88.5**—**85.2** | **96.2** | 95.5 |
| | Column-synonym | **71.0**—**57.2** | 68.4—54.7 | **77.7** | 76.4 |
| | Column-carrier | **86.9**—**64.9** | 80.8—51.1 | **73.0** | 60.9 |
| | Column-attribute | 58.8—**56.3** | **68.9**—46.2 | **82.9** | 64.6 |
| | Column-value | 82.9—69.4 | **86.2**—**71.4** | **81.7** | 79.8 |
| | value-synonym | **72.5**—53.0 | 72.1—**59.9** | 69.2 | **78.6** |
| | Multitype | **74.4**—**57.1** | 72.6—53.7 | **72.8** | 70.7 |
| | Others | **79.6**—**78.3** | 76.6—69.7 | **96.1** | 87.1 |
| | Average | **76.0**—**65.0** | 75.3—60.8 | **82.0** | 77.3 |
| SQL | Comparison | 68.0—**68.0** | **73.6**—66.9 | **89.3** | 84.7 |
| | Sort-order | **79.2**—**74.5** | 57.8—57.8 | **92.1** | 87.4 |
| | NonDB-number | 83.2—77.1 | **87.8**—**89.3** | 92.7 | **100.0** |
| | DB-text | 64.7—65.1 | **73.3**—**72.4** | **94.1** | 89.7 |
| | DB-number | **86.3**—**85.1** | 80.5—79.3 | **98.3** | 96.1 |
| | Average | **76.3**—**74.0** | 74.6—73.1 | **93.3** | 91.6 |
| All | | **76.6**—**65.9** | 74.6—64.4 | **83.0** | 81.9 |

Table 16: The execution (EX) accuracy of SOTA finetuned and in-context-learning text-to-SQL models on the original Spider development set (Spider-dev) and Dr.Spider. The first two result columns are pre-perturbation and pose-perturbation accuracy of PICARD and CODEX, where x—y represents the pre-perturbation accuracy and post-perturbation accuracy (i.e. absolute robustness accuracy). The last two result columns are the relative robustness accuracy of PICARD and CODEX. We report macro-average scores over multiple perturbations. Bold number is the highest accuracy in each category.

