# OpenReview forum: "Dr.Spider: A Diagnostic Evaluation Benchmark towards Text-to-SQL Robustness"
_ICLR.cc/2023/Conference — ICLR 2023 notable top 5%_

### Official Review · Reviewer_MJgi · 2022-10-21

**Confidence:** 4
**Correctness:** 4
**Technical Novelty And Significance:** 3
**Empirical Novelty And Significance:** 3
**Recommendation:** 8

**Clarity, Quality, Novelty And Reproducibility:**

Clarity: good. The Reviewer had no trouble following the main idea and details.

Quality: good. The paper addressed a clearly defined problem and the evaluations supported the claim.

Novelty: ok. The methodology employed are mostly known in the field. As pointed out in the strength&weakness, the cost/benefit of using  PLM is not very clear.

Reproducibility: good. The authors promised to release the data and tools.

**Strength And Weaknesses:**

Strength:

1) A clear and useful goal. The paper addresses a specific and useful problem of evaluating the robustness of text-to-SQL systems, where existing datasets either cannot do, or have gaps in coverage.
2) Detailed and useful evaluation. Most of STOA algorithms are evaluated, with detailed break-downs and also examples to help reader understand.
3) Clear writing. The paper is well written, The Reviewer had no trouble following the main idea and technical details.
4) Helpful to the community. The dataset will be released, which would help the improvement of the robustness of related methods.

Weakness:

1) The necessity and complexity of using PLMs are not clear, as 1) the total amount of data generated is small, in the range of hundreds. 2) significant human review is still needed up and down the pipeline.
2) The size of DR SPIDER (The Reviewer's impression is ~1k) is much smaller than SPIDER (~10K questions).


Detailed suggestions:

1) Section 2 paragraph 1, last sentence: it's better to explain why we need semantic-changing perturbations, e.g. to be used as negative examples? or to test the model's ability to distinguish between closely related but different semantics?
2) Section 3.2 paragraph 2: better explain what is "naturally occuring  tables and columns".
3) Last paragraph in Section 3.2: Why split the filtered data into chunks, while ensuring each chuck have one annotators, why not just annotate all without chuck-splitting?
4) Section 4. Upper-case of common model names, "Bert-large" -> "BERT-large" etc.


**Summary Of The Paper:**

The paper created a dataset to evaluate text-to-SQL systems on their robustness. The dataset, DR.SPIDER, is an extension of the openly available SPIDER dataset. It added perturbation to the natural language questions, SQL statements and example database (schema and values), so more robust text-to-SQL systems would have less degradation in accuracy on the perturbed samples. The paper categorized the perturbations into 17 different categories, covering different aspects of possible real-world variations.

To obtain the perturbations, the paper employed multiple methods. It used synonym sets from previous related works, employed in-house domain experts, used online crowd-source workers (Amazon Mechanical Turks), and leveraged PLMs. The process focused more on getting a balanced and correct dataset than the effectiveness of any single method.

The paper performed extensive evaluations of state-of-the-art models on DR.SPIDER, showed the ability of robustness measurement of the dataset, and also provided insights on the factors that may affect the robustness of existing algorithms.

The authors promised to release the dataset and evaluation tools.

**Summary Of The Review:**

The paper provides a good extension to the SPIDER dataset to measure the robustness of text-to-SQL models. It addresses a clearly defined and important aspect of evaluation. The construction of the dataset is well thought, and the effectiveness of it has been demonstrated with experiments involves SOTA models. The paper is also clearly written and easy to follow. The resulted dataset could be a useful resource for the community in the future.

The only concern is the size of the set (a magnitude smaller than SPIDER, if the Reviewer understood correctly). The final size of the set is not clearly mentioned in the abstract, intro or conclusion (only mentioned "17 perturbations").

---

> ### Author Response · Authors · 2022-11-16
> **Response to Reviewer MJgi**
>
> Thank you for the appreciation of our work and valuable suggestions!
> To answer your specific questions:
>
> >The necessity and complexity of using PLMs are not clear, as 1) the total amount of data generated is small, in the range of hundreds. 2) significant human review is still needed up and down the pipeline.
>
> We use PLMs for their scalability and the capability of few-shot learning. 1) Compared to human annotations, PLMs with in-context learning are more scalable. For example, the most powerful GPT-3 model costs $0.02 per 1k tokens, which is significantly lower than the cost of obtaining paraphrases from human annotators. 2) We aim at evaluating text-to-SQL models on text-specific paraphrases, which makes most finetuning-based paraphrase models not applicable as the annotated paraphrases from crowdsourcers are not enough for training or finetuning a paraphrasing model. However, PLMs with in-context learning show strong few-shot learning ability in learning paraphrasing in text-specific categories.
> We agree that human review is still required to guarantee a high quality of our data. However, the effort of detecting if a sentence is a paraphrase of another is much lighter than writing a paraphrase. Moreover, we would state that such reviewing stage is necessary since even the crowdsourcing human paraphrase only gives 54% of paraphrases factual. The text-to-SQL task is to translate a natural language question to a SQL query where a slight change on almost any part of the natural language question may alter its semantics (corresponding SQL), so we believe that such human reviewing is necessary for high-quality data.
>
> >The only concern is the size of the set (a magnitude smaller than SPIDER, if the Reviewer understood correctly). The final size of the set is not clearly mentioned in the abstract, intro or conclusion (only mentioned "17 perturbations").
>
> Dr. Spider contains 15K pairs of pre-perturbation and post-perturbation examples, which is much larger than the Spider development set (1K), as one example can be perturbed in multiple ways and therefore fell into multiple categories. We added the size of Dr. Spider to the Introduction and the beginning of the Data Creation section in the revision.
>
>
> >Section 2 paragraph 1, last sentence: it's better to explain why we need semantic-changing perturbations, e.g. to be used as negative examples? or to test the model's ability to distinguish between closely related but different semantics?
>
> We use semantic-changing perturbations to evaluate models’ decision boundaries on closely related but different semantics. We added a clarification to the revision.
>
> >Section 3.2 paragraph 2: better explain what is "naturally occuring tables and columns".
>
> We use the term "naturally occurring" to refer to the tables and columns that are not necessary for understanding questions to the database. We omit the description of these tables/columns as they make annotators tend to follow the schema during paraphrasing. For example, “name” and “grade” are naturally occurring columns for table “student”. A crowdsourcer can understand the question “Which grade is Alice in?” as long as they know the database is about students. Without knowing the column “grade” in the database, a crowdsourcer is more likely to paraphrase the question without following the database schema, such as “What year is Alice in?”. We added a clarification to the revision.
>
> > Last paragraph in Section 3.2: Why split the filtered data into chunks, while ensuring each chuck have one annotators, why not just annotate all without chuck-splitting?
>
> The annotators here are SQL experts and trained with examples of each paraphrase category first. We then sampled 5% paraphrases and required all annotators to review them. The Fleiss’ kappa is 0.61 which shows a substantial agreement. Therefore, to reduce the reviewing cost, we split the rest data into chunks and required each chunk to be reviewed by one annotator.

---

### Official Review · Reviewer_WKcN · 2022-10-22

**Confidence:** 2
**Correctness:** 4
**Technical Novelty And Significance:** 2
**Empirical Novelty And Significance:** 4
**Recommendation:** 8

**Clarity, Quality, Novelty And Reproducibility:**

I found this work to be straightforward in a good way: they present some novel ideas for generating more meaningful variation in benchmark data which are informed by a good understanding of the problem and existing approaches. They generate this new dataset carefully and systematically, show how existing methods struggle with the new variation, and do some analysis to suggest future directions. There are prior robustness-oriented extensions in this domain, but this work seems to go further in a meaningful way. The work seems very well-executed and clearly described.

Table 3: DB Perturbation "Average" is misspelled as "Avergae"

**Strength And Weaknesses:**

The paper is clearly written, systematically laying out the kind of data variation they are targeting and how they aim to achieve it. The experimental results show significant problems or gaps with existing models against these variations, but the error analysis also points to potential directions for improvement. The paper seems to reflect deep domain knowledge on this problem area and good contextualization with respect to related works.

I found the paraphrase categories (Table 1) very helpful.

The Appendix is extensive, containing fairly fine-grained details about the implementation and execution of the experiments, as well as even more details on the results themselves. To be honest I think the Prompt Prefix tables (Appendix C) could be put in the code repo instead but it is a minor suggestion.

The error analysis and diagnostic insights seem very promising for future work. It's interesting that PICARD got confused by "8 youngest winners" but I guess queries in datasets probably often ask for Top 3 vs Top 8.

Fleiss Kappa of 0.61 is reasonable but not perfect agreement - were there interesting patterns of disagreement among annotators?

There may be weaknesses that a deeper expert on this area may have better context on, but I saw no red or yellow flags.

**Summary Of The Paper:**

This work contributes a robustness benchmark built on top of the Spider text-to-SQL benchmark dataset. One novel aspect is the use of crowdsourcing in tandem with language models to generate paraphrases of the natural language queries. The paper presents an evaluation of recent models against this new benchmark along with a corresponding error analysis.

**Summary Of The Review:**

This seems like a valuable contribution to this problem area that should spur further research and improvements.

---

> ### Author Response · Authors · 2022-11-16
> **Response to Reviewer WKcN**
>
> Thank you for the detailed review and insightful feedback!
>
> >The error analysis and diagnostic insights seem very promising for future work. It's interesting that PICARD got confused by "8 youngest winners" but I guess queries in datasets probably often ask for Top 3 vs Top 8.
>
> We also find the example particularly interesting. We did find the training data asks for the Top 3 more often than the Top 8. A robust model should understand the number here corresponds to the LIMIT clause and the value of the number should affect the SQL structure prediction, but Picard fails to maintain the correct SQL structure. More interestingly, we found GPT-3 CodeX davinci (trained on a large amount of GitHub code) is very robust to NonDB number. We believe that the diverse training code with real-world distribution (codes on GitHub) improves model robustness to it.
>
> >Fleiss Kappa of 0.61 is reasonable but not perfect agreement - were there interesting patterns of disagreement among annotators?
>
> After taking a deeper look at the disagreement annotations, we find that even though annotators are trained with the definition and examples of each paraphrase category, they may have different “decision boundaries” for those categories. An interesting phenomenon is that different annotators believe a paraphrase should belong to different categories. For example, “Return the template type code of the template that is used by a document named Data base.” is paraphrased to “What is the template type code of the template that is used by the Data base document?” where “Data base” is a cell value under column “document name”. The corresponding database contains a table with columns “document id”, “document name”, and “document details”. Two annotators believe the paraphrase falls into the Value_synonym category as the value “Data base” indicates the column “document name” instead of other columns, while the other annotator disagrees as the word “document” still exists after the paraphrase.  We believe it is acceptable during annotation as natural language is complex and hard to be divided into categories exclusively. That’s why we want to include Multitype and Others categories for paraphrasing.
>
> > Prompt Prefix tables (Appendix C) could be put in the code repo instead. "Average" is misspelled as "Avergae"
>
> We appreciate the suggestion and we will move the prompt tables in Appendix C to the code repo to make the paper more concise when we release data and code. Also, we corrected the misspelling in the revision.

---

### Official Review · Reviewer_QnJr · 2022-10-24

**Confidence:** 4
**Correctness:** 4
**Technical Novelty And Significance:** 3
**Empirical Novelty And Significance:** 3
**Recommendation:** 8

**Clarity, Quality, Novelty And Reproducibility:**

* The paper is well-written and easy to follow.
* Figure 1 and Table 1 are great at helping readers follow along with the particularities of Dr. Spider.
* Augmenting datasets and benchmarks is not a new idea, but, as the experiments show, it is necessary. I think that applying a known approach where necessary is a positive.
* The paper is clear on how the data was collected and the appendices add all necessary details to understand the data collection.


**Strength And Weaknesses:**

## Strengths

+ The paper is well-written and easy to follow.
+ It is important that SQL benchmarks test for linguistic variations and challenge systems to do more than lexical matching between the NLQs and the SQL queries
+ The experiments are convincing and thorough.
+ Figure 1 and Table 1 are great at helping readers follow along with the particularities of Dr. Spider.

## Weaknesses

- Did not see any.


**Summary Of The Paper:**

The paper introduces a variation of the SPIDER evaluation benchmark called Dr. Spider. Dr. Spider with the dr standing for Diagnostic Robustness, improves the original SPIDER benchmark by introducing augmentations to the Natural Language Quries and the SQL. While the augmentations do not change the benchmark, the challenge systems' understanding of natural language, and SQL. The augmentations aim to reveal whether models are overfitting to SPIDER's perks and to challenge systems to generalize better. The paper's experiments show that all state-of-the-art systems score lower on Dr. Spider, thus proving the need for an augmented SPIDER.

**Summary Of The Review:**

This is a well-written paper with no major weaknesses.

---

> ### Author Response · Authors · 2022-11-16
> **Response to Reviewer QnJr**
>
> Thank you for your positive feedback and appreciation of our work!
>
> We are glad to see that you found our data collection details are presented clearly and Figure 1 and Table 1 are helpful.

---

### Official Review · Reviewer_gzxj · 2022-10-30

**Confidence:** 3
**Correctness:** 3
**Technical Novelty And Significance:** 3
**Empirical Novelty And Significance:** 3
**Recommendation:** 8

**Clarity, Quality, Novelty And Reproducibility:**

Clarity: The paper is well written and easy to follow

Quality: I believe the proposed dataset will be of high quality and would serve as an interesting benchmark to test the robustness of text-to-sql models

Novelty: Even though robustness of text-to-sql models were studied before, the paper does a more extensive and thorough job of creating perturbations. Hence, I believe the paper still makes a novel contribution

Reproducibility: The dataset will be released to the public, so yes.

**Strength And Weaknesses:**

**Strengths**

- Developing robust text-to-SQL models is important and this benchmark can serve an important role to test that
- The paper covers more comprehensive perturbation for all components of text-to-SQL tasks. Moreover the use of LLMs to generate perturbation makes their process scalable without making the task artificial (e.g. same fluency scores of questions)
- The paper is clearly written and easy to understand

**Weaknesses**

- I would have liked to see the robustness performance of large LMs such as GPT-3. It would be interesting to see how much incontext learning with large LMs can be robust to the proposed perturbations

**Summary Of The Paper:**

This paper proposes a benchmark to test the robustness of text-to-SQL models. Specifically, the benchmark is created to test whether such models work well (i) when names of particular columns in a table are replaced with synonymous strings (DB perturbations), (ii) when natural language questions are replaced by its paraphrases (NLQ perturbation) and (iii) when minor changes are in the natural language query and the SQL query (SQL perturbation). The paper extends the Spider dataset (Yu et al 2018) to cover all the perturbations. They propose a total of 17 perturbations belonging to these 3 categories.

DB perturbation: For DB perturbation the paper proposes replacing column names by synonyms and abbreviations. Moreover, they also propose a novel perturbation where they replace a column by one-or-more multiple equivalent semantic columns (e.g. the “name” column can be replaced by two columns “first_name”, “last_name” etc)

NLQ perturbation: For NLQ perturbation, the paper proposes a scalable approach in which first crowd workers propose 5 paraphrases of 100 questions. This is followed by experts filtering and categorizing the paraphrases into 9 categories. Next, to scale the process, an LLM is prompted with paraphrases from each category to generate paraphrases of new questions. This is followed by another automated filtering stage where an NLI model is used followed by the final round of expert filtering.

SQL perturbation: For SQL perturbation, they replace operations like comparisons, sort-order, etc from both the natural language and SQL query.

The paper tests a variety of text-to-SQL models on this dataset and finds that the most robust models suffer from a 14% drop in performance with around a 50.7% drop on the most challenging perturbation.

**Summary Of The Review:**

I enjoyed reading the paper and I believe this will be an interesting and strong benchmark. I am leaning toward accepting the paper.

---

> ### Author Response · Authors · 2022-11-16
> **Response to Reviewer gzxj**
>
> Thank you for the clear summary of our paper and the insightful suggestion!
>
> > I would have liked to see the robustness performance of large LMs such as GPT-3. It would be interesting to see how much in-context learning with large LMs can be robust to the proposed perturbations
>
>
> We evaluate the robustness of a large LM, GPT-3 CodeX [1] for in-context learning on Dr. Spider. The table below summarizes the results of CodeX and Picard.
> We use GPT-3 CodeX davinci with the Create Table + Select 3 prompt from [2], as it achieves the best performance with in-context learning (74.0 execution accuracy on the original Spider development set based on our implementation), which is even comparable with the SOTA finetuned model, Picard (79.3 execution accuracy). We have a few interesting findings (1) In-context CodeX is more robust than finetuned Picard to database perturbations even though it still suffers a performance drop from 72.6 to 60.7, and (2) In-context CodeX doesn’t show better robustness against NLQ perturbations than finetuned Picard.
>
> We hypothesize that the robustness of CodeX is largely related to its training data. As the training data are obtained from GitHub codes, CodeX has seen various distributions of databases. Specifically, we find that CodeX suffers a limited performance drop to Schema-abbreviation perturbation (from 70.2 to 68.6), compared to Picard (from 74.9 to 64.7). We believe that CodeX is robust to the Schema-abbreviation perturbation, as it has seen a large number of abbreviation schemas in its training data.
>
> In NLQ perturbations, we find two categories that are particularly challenging to CodeX: Column-carrier (from 80.8 to 51.1), and Column-attribute (from 68.9 to 46.2). Those perturbations aim at implicitly mentioning a column with the question carrier or the attribute of the column. For example, In Table 1, the column “name” in “Show the name of teachers” is implied by “Which teachers”, and the column “year” in “worked the greatest number of year” is implied by “worked the longest”. The codeX was trained with the goal of generating code from docstrings. The language style of common docstrings is slightly different from the style of spoken language therefore CodeX may not be familiar with such implicit mentions. As the CodeX details are not fully published, the analysis is subjective.
>
> We added the results and analysis to the appendix in the revision.
>
> | Model | Spider-dev | DB perturbations | NLQ perturbations | SQL perturbations | Dr. Spider Overall|
> | ----------- | ----------- | ----------- | ----------- | ----------- | ----------- |
> | Picard | 79.3 | 78.9 —55.0| 76.0 —65.0| 76.3 —74.0| 76.6 —65.9|
> | CodeX | 74.0 |72.6 —60.7| 75.3 —60.8| 74.6 —73.1| 74.5 —64.4|
>
>
> [1] Evaluating Large Language Models Trained on Code https://arxiv.org/pdf/2107.03374.pdf
>
> [2] Evaluating the Text-to-SQL Capabilities of Large Language Models https://arxiv.org/pdf/2204.00498.pdf

---

### Author Response · Authors · 2022-11-16
**Rebuttal Revision**

We thank all reviewers for their positive feedback and valuable suggestions. We appreciate that reviewers acknowledge that:
1. Our benchmark can serve an important role to test the robustness of text-to-SQL models and spur further research. (R1,2,3,4)
2. Our analysis is insightful. (R2,3,4)
3. The use of crowdsourcing in tandem with language models to generate paraphrases of natural language queries is novel. (R3)
4. The paper is clearly written. (R1,2,3,4)

We have revised our draft based on the suggestions from reviewers. We summarize the updates below:
1. We added a study of in-context learning models (GPT3 CodeX) robustness to Appendix B. (R1)
2. We added the size of Dr.Spider in the Introduction (Section 1) and the beginning of the Data Creation (Section 3). (R4)
3. We elaborated on “semantic-changing perturbations” (Section 2) and “natural occurring” (Section 3.2), and corrected misspellings. (R3, R4)

---

### Decision · Program_Chairs · 2023-01-20

**Decision:**

Accept: notable-top-5%

**Justification For Why Not Higher Score:**

N/A

**Justification For Why Not Lower Score:**

The paper presents a new benchmark and is well-written. The paper received good scores from all reviewers, and the authors did a good job addressing the reviewers' comments.

**Metareview: Summary, Strengths And Weaknesses:**

The paper presents a new benchmark, Dr.Spider, based on the Spider dataset for evaluating text-to-SQL models. Dr.Spider contains a set of perturbations on databases, natural language questions, and SQL queries to measure the robustness of the models against those perturbations. The paper also argues the necessity of developing the new benchmark and supports its arguments by demonstrating the performance loss of the SOTA models on Dr.Spider due to the perturbations compared to the Spider dataset. The authors also did an excellent job addressing the reviewers' comments in the rebuttal. The new benchmark will certainly support the developments in neural text-to-SQL research.

**Note From Pc:**

if the above contains the word "oral" or "spotlight" please see: "oral" presentation means -> notable-top-5% and "spotlight" means -> notable-top-25%. As stated in our emails, we are disassociating presentation type from AC recommendations